# Functional, Physical, and Volatile Characterization of Chitosan/Starch Food Films Functionalized with Mango Leaf Extract

**DOI:** 10.3390/foods12152977

**Published:** 2023-08-07

**Authors:** Cristina Cejudo, Marta Ferreiro, Irene Romera, Lourdes Casas, Casimiro Mantell

**Affiliations:** 1Chemical Engineering and Food Technology Department, Wine and Agrifood Research Institute (IVAGRO), University of Cadiz, Avda. República Saharaui, s/n, 11510 Cadiz, Spain; cristina.cejudo@uca.es (C.C.); irene.romeragonzalez@alum.uca.es (I.R.); casimiro.mantell@uca.es (C.M.); 2Analytical Chemistry Department, Wine and Agrifood Research Institute (IVAGRO), University of Cadiz, Avda. República Saharaui, s/n, 11510 Cadiz, Spain

**Keywords:** food packaging, *Mangifera indica*, food preservation, Volatile Organic Compounds, Sustainable Developments Goals

## Abstract

Active packaging is one of the currently thriving methods to preserve highly perishable foods. Nonetheless, the integration of active substances into the formulation of the packaging may alter their properties—particularly mass transfer properties—and therefore, the active compounds acting. Different formulations of chitosan (CH), starch (ST), and their blends (CH-ST), with the addition of mango leaf extract (MLE) have been polymerized by casting to evaluate their food preservation efficiency. A CH-ST blend with 3% MLE using 7.5 mL of the filmogenic solution proved to be the most effective formulation because of its high bioactivity (ca. 80% and 74% of inhibition growth of *S. aureus* and *E. coli*, respectively, and 40% antioxidant capacity). The formulation reduced the water solubility and water vapor permeability while increasing UV protection, properties that provide a better preservation of raspberry fruit after 13 days than the control. Moreover, a novel method of Headspace-Gas Chromatography-Ion Mobility Spectrometry to analyze the volatile profiles of the films is employed, to study the potential modification of the food in contact with the active film. These migrated compounds were shown to be closely related to both the mango extract additions and the film’s formulation themselves, showing different fingerprints depending on the film.

## 1. Introduction

The reduction in food loss and waste are of primary importance to guarantee the food supply worldwide, so it has been included as one of the goals to achieve among the Sustainable Developments Goals for 2030. In this sense, the efficient use of preservative techniques plays an essential role in the food industry since it can extend the shelf life of food products. The preservation of food products facilitates their handling and transport, which allows the consumption of foodstuffs in the locations where they are not produced, thus ensuring foreign trade of fresh or minimally processed products, as fourth-range products based on fruit and vegetables. However, the shelf life of these products is quite short, and they are particularly susceptible to deteriorating agents [1]. Oxidation and hydrolysis of lipids are some of the chemical alterations that food can undergo and that may cause rancidity and off-flavor, enzymatic degradations associated with color and texture alterations, non-enzymatic browning due to Maillard and hydrolysis reactions, as well as the growth of bacteria, yeasts, and/or molds [2]. In recent years, fresh food packaging is focused on increasing the use of biodegradable bio-based polymers, which already account for about 36.3% of industrially produced bioplastics [3]. In this sense, the studies of film biodegradable formulations, and their effectiveness in real food experiments, have yet to be studied to achieve industrial applications and thus, lead to higher implementation of biodegradable films in the market.

In the biodegradable polymer field, starch is one of the products that can be easily and cheaply obtained from agricultural by-products [4]. Nevertheless, naturally-based film packaging usually presents unsatisfactory mechanical, thermal, or water barrier capacities when compared to more conventional packaging materials [5]. In the case of starch, undesirable changes in the thermomechanical properties caused by the re-crystallization and retrogradation of its structure can jeopardize its use as the only polymer in the film formulation [6,7]. Furthermore, chitosan stands out because of its intrinsic antioxidant and antimicrobial activity, film-forming capacity, and good permeability to CO_2_ and to O_2_ [8], and it has been widely combined with starch-based polymers to enhance their properties [9,10,11], together with the incorporation of cross-linkers, surfactants, and even active substances that intervene in food preservation [4,12]. This last option is very convenient since it also provides functional properties, turning into active packaging materials. In this matter, the use of natural substances from agri-food by-products is an ongoing trend, because it improves the added value of the film and promotes the food industry to its transformation towards a circular economy model. The use of essential oils from aromatic herbs and spices, such as thyme, eucalyptus, parsley, or rosemary [13,14,15], or natural extracts, rich in phenolic compounds, obtained from agricultural by-products, such as pomaces [16,17,18] or leaves [19,20,21], have been reported. Mango-derived compounds have been previously added to films, both as nutraceuticals [22] and food preservers [23,24,25], with promising results in terms of the transferring of their properties to the packaging polymer film. These additives have the aim of modifying the film’s permeability to certain gases, thus preventing the growth of microorganisms, retarding food oxidation, or even slowing down the ripening of packed fruits or vegetables [26].

Despite the advantages of functionalized polymers for food packaging, the migration of active compounds from the packaging to the food is one of the key aspects of the development of these technologies. In fact, not only the polymer/active substance formulation and their interactions determine their effectiveness, but also the technique used for the production of the films can have an influence on the migration of the active compounds [27]. One of the drawbacks associated with the use of natural additives is that high concentrations are to be added to the polymer film formulation to be effective, which could result in the transfer of undesirable odors and flavors to the food [26]. In this sense, the possible packaging/food migration that may occur must be monitored and controlled. Until now, the majority of the analytical techniques used to corroborate the kinetic release of compounds from active films have focused on the identification of non-volatile compounds, mainly polyphenols, to assess their bioactivity. These studies are usually performed in food simulants or buffer solutions further analyzed spectrophotometrically or by liquid chromatography [28,29,30]. However, in food-contact materials, it is also important to corroborate if any Volatile Organic Compounds (VOCs) are released from the biofilms into the headspace of the food package. The migration of these VOCs might either promote preservation and/or modify the organoleptic properties of the food. The conventional methods used to detect VOCs from food packaging are generally time-consuming, non-ecological, and require skilled labor [31]. However, in recent years, the potential of Headspace-Gas Chromatography-Ion Mobility Spectrometry (HS-GC-IMS) is emerging in the food control field for the analysis of VOCs, because of its numerous advantages [32] such as short time analysis, ultrahigh sensitivity, high stability, and good repeatability. HS-GC-IMS has been successfully applied to determine VOC changes for the rapid detection of contamination in certain foods, such as corn or wheat [33,34], although the use of this technique to assess the volatile migration from the active film has not been previously tested.

For all the above said, this work intends to determine the influence of the formulation of starch and chitosan biodegradable active films containing mango leaf extract on its physical, functional, and preservative properties, as well as to evaluate the migration of volatile compounds into the headspace of the package depending on their different formulation, in order to establish the most suitable one. The selected films were subsequently used to preserve raspberries as an example of a highly perishable fresh product. The preservation has been visually monitored for a 13-day storage period.

## 2. Materials and Methods

### 2.1. Chemicals and Raw Materials

The *Mangifera indica* L. leaves, Kent variety, were supplied by the Institute of Subtropical and Mediterranean Horticulture “La Mayora” from the Spanish National Research Council (CSIC) (Malaga, Spain) and used as the raw material for the extraction process. For the extractions, CO_2_ (99.99% purity) supplied by Abelló Linde (Barcelona, Spain) and ethanol supplied by Panreac (Barcelona, Spain) were used. For the development of the films, chitosan (medium molecular weight, ≥75% deacetylated degree) was supplied by Sigma-Aldrich (Steinheim, Germany), and starch (with 20 to 25% amylose content), acetic acid, and glycerol supplied by Panreac (Barcelona, Spain) were employed. The following reagents, also from Sigma-Aldrich (Steinheim, Germany), were used: 2,2-Diphenyl-1-picrylhydrazyl (DPPH) was employed to evaluate the antioxidant capacity, and 2,3,5-triphenyl tetrazolium chloride (TTC) was used as a colorimetric marker for its antimicrobial activity. Peptone, yeast extract and sodium chloride (NaCl) were used to prepare the Luria Bertani (LB) growth medium. The antimicrobial capacity assays were carried out using pathogenic *Escherichia coli* (CECT101) and *Staphylococcus aureus* (ATCC 6538) bacteria from the Spanish Type Culture Collection (CECT, Valencia, Spain). The raspberries used for the preservation experiment were purchased from a local market.

### 2.2. Supercritical Extraction of Mango Leaf Extract (MLE)

The Mango Leaf Extract (MLE) was obtained by enhanced solvent extraction (CO_2_ + ethanol 1:1) at 250 bar and 80 °C by batch mode in an SF1000 equipment (Thar Technologies, Pittsburg, PA, USA), following the method described by Rosales et al. [35]. Approximately 500 g of crushed dried leaves were used in the extraction process, obtaining an extract with a final concentration of 91.8 mg/mL that was used for the film manufacture.

### 2.3. Development of Bioactive Films

The films made of chitosan (CH), starch (ST), or a combination of both compounds (CH-ST) were produced by solvent casting in 55 mm-diameter polystyrene Petri dishes. The chitosan films were prepared using the method described by Siripatrawan and Harte [36] and Ruiz-Navajas et al. [37], with some modifications. A 2% *w*/*v* chitosan solution was prepared with distilled water containing 1% *w*/*v* acetic acid and 0.3% *v/v* glycerol (pH 2.53). This solution was shaken vigorously for 15 min on a magnetic plate to favor the homogenization of the components. After this, a specific volume of extract was added to the solution and the shaking was performed another 15 min. Then, the preparations were subjected to ultrasound and vacuum for 15 and 35 min, respectively, to remove any trace of air from the solution that might interfere with the film polymerization. Finally, the different volumes were poured into Petri dishes and allowed to dry at room temperature for 48 h. On the other hand, the starch films were produced based on the procedures described by Piñeros-Hernández et al. [38] and Reddy and Yang [39]. The solutions were composed of 2% *w*/*v* starch and 2.4% *v*/*v* glycerol in distilled water (pH 6). The preparations were first heated at 85 °C under stirring. After tempering, a certain volume of MLE was added and shaken for 15 min. Then, the solutions were plated and polymerized for 72 h at room temperature. Finally, a chitosan-starch polymer was also produced following the procedure employed by Lozano-Navarro et al. [40]. The preparation of the 2% *w*/*v* chitosan (1% *w*/*v* acetic acid) and the 2% *w*/*v* starch solutions were carried out separately according to their respective protocols. Then, both suspensions were mixed together at 1:1 volume ratio after the starch suspension was tempered (pH 2.67). Glycerol 0.25% *v/v* was added and the mixture was subjected to agitation for 30 min, adding the extract halfway this time. The resulting solution was treated for 15 min with ultrasounds and kept for 35 min under vacuum before being poured into the Petri dishes. The drying of the films was carried out for 72 h at room temperature.

The volume of the filmogenic solution (5, 7.5, and 10 mL) and the MLE amount (1, 3, and 5% *v*/*v*) to be used for the film processing were investigated. Each formulation was manufactured at least four times to obtain enough films to carry out all the experiments. Those formulations were studied in order to determine the optimal composition, given that both the polymer solution volume and the amount of extract are important factors with regard to certain characteristics of the resulting film, such as gas and water vapor permeability, transparency, and other mechanical properties [25]. Moreover, the film thickness, which is largely dependent on the volume of liquid solution used, can have an impact on its mechanical strength, extract holding capacity, water vapor permeability, and transparency. Some of those parameters have been determined according to the procedure described in the following sections.

### 2.4. Antioxidant Capacity of the MLE and Bioactive Films

The antioxidant capacity of the extract was determined following the procedures described by Cejudo et al. [41], by quantifying the reduction of the free radical 2,2-Diphenyl-1-picrylhydrazyl (DPPH) in the presence of an antioxidant compound. The MLE antioxidant capacity curve was prepared using different concentrations of the extract in ethanol: 2000, 1000, 500, 250, 125, and 62.5 μg/mL. Aliquots of 0.1 mL from the different concentrations of extract were poured into 3.9 mL of 6 × 10^−5^ M DPPH ethanolic base and kept in the absence of light for 2 h. The absorbance of each sample was measured at 515 nm. The analyses were performed in duplicate. By relating the percentage of remaining DPPH vs. the concentration of the MLE, Equation (1) was obtained and *IC*_50_ was calculated accordingly. Then, the Antioxidant Activity Index (*AAI*) was also calculated following Equation (2).
(1)$$\%\;\mathrm{remaining}\;\mathrm{DPPH}\;=0.0994C^{2}+6.0404C+99.779\%\;R^{2}:0.9973$$
(2)$$AAI=\frac{C_{DPPH\;i}}{IC_{50}}$$

In the case of solid samples as the bioactive films, 15 mg of film were submerged into 4 mL of the radical solution of DPPH for 6 h to allow the MLE to diffuse into the reagent medium. The variation of the absorbance at 515 nm was monitored, and the %I was calculated according to Equation (3). During this time, the samples were kept in the absence of light and at room temperature. The tests were performed in duplicate.
(3)$$\%I=\left(1-\frac{Abs_{f}}{Abs_{i}}\right)\times 100$$
where *Abs_f_* is the absorbance of the samples and *Abs_i_* is the initial absorbance of the DPPH reagent.

### 2.5. Antimicrobial Capacity of the MLE and the Bioactive Films

The antibacterial capacity of the MLE was quantitatively determined by microdilution in a liquid medium using the 2,3,5-triphenyltetrazolium chloride (TTC) reagent as the colorimetric marker, according to the procedure described by Cejudo Bastante et al. [42]. TTC is reduced as a function of cell viability so that it turns red in the presence of viable cells [43]. The study was performed using Escherichia coli (Gram-negative) and Staphylococcus aureus (Gram-positive), to assess the effectiveness of the extract against different bacteria. Initially, a number of extract dilutions were prepared at concentrations of 93.75, 187.5, 375, 750, 1500, and 3000 μg/mL. Also, a bacterial inoculum in an LB medium of 1.5 × 10^6^ CFU/mL was prepared and 100 μL of the same was poured into each well together with 10 μL of the extract at different concentrations. After 24 h of incubation at 35 °C, 10 μL of TTC at 5 mg/mL was added to each well to analyze the cell viability. Then, the plate was incubated for an additional 30 min under the same conditions. The changes in color were measured by means of a Synergy™ HTX Multi-Modal Microplate Reader spectrophotometer (BioTek Instruments, Winooski, VT, USA) at 500 nm. The determinations were performed in triplicate and a blank of the extract was used as reference in order to discard any interference. In addition, a positive control test was employed to verify the actual growth of the inoculum. For the quantification of the antibacterial capacity of the extract, the percentage of inhibition of the microbial growth (%*IMG*) was calculated according to Equation (4).
(4)$$\%IMG=\left(1-\frac{Abs_{f}}{Abs_{c}}\right)\times 100$$
where *Abs_f_* is the absorbance of the samples and *Abs_c_* is the absorbance of the positive control.

During the antioxidant activity tests, the ability of the phenolic compounds to diffuse through the matrix into the medium was evidenced. Therefore, the antimicrobial activity of the films with the best antioxidant results was analyzed. For this purpose, 50 mg of film were placed into sterilized tubes containing 10 mL of LB medium and were let to diffuse for 24 h at 35 °C prior to adding the microorganism. As a positive control, tubes with film without extract addition were prepared to evaluate the normal growth of the strains. Then, the absorbance of each sample was measured at 625 nm to obtain their respective blanks. After that, the samples were inoculated with the microorganisms at a concentration of 1.5 × 10^6^ CFU/mL, and the samples were again incubated. Their absorbance was measured after 24 h and the cell concentration in each sample (*Cc* (UFC/mL)) was calculated following Equation (5), obtained by the McFarland Standards (0.5 to 4) at 625 nm:*Cc* = 14.445·Δ*Abs* − 0.0962(5)
where Δ*Abs* is the absorbance difference between the absorbance of each sample after the incubation time and its respective blank. Finally, the percentage of inhibition of microbial growth was calculated as expressed in Equation (6):(6)$$\%IMG=\left(1-\frac{C_{cf}}{C_{ci}}\right)*100$$
where *C_cf_* is the final cell concentration found from each tube and *C_ci_* that of the positive control.

### 2.6. Optical Properties of the Films

Color and opacity of food packaging are features of great importance in the perception of consumers, who prefer transparent packages that allow them to be aware of the state of the food inside [44,45]. The color of all the films produced was measured by means of a colorimeter, using the CIELAB scale. The parameters *L** (black-white), *a** (green-red), and *b** (blue-yellow) were determined. The variable *L** corresponding to luminosity is indicated along the vertical plane using values from 0 to 100, where 0 corresponds to white and 100 to black. The positive values of the variable *a** indicate red tones, while its negative values denote green ones. And finally, the positive values of *b** correlated with the yellow tones of the sample, while its negative values indicate the blue tones. Measurements were carried out using a portable spectrophotometer (CM-2600d, Konica Minolta Co., Osaka, Japan) following the method described by Cejudo et al. [46]. In addition, in order to establish the total color difference between the functionalized films and their corresponding control samples, the Δ*E* index, calculated according to Equation (7):(7)$$\Delta E=\sqrt{{\left(\Delta L^{*}\right)}^{2}+{\left(\Delta a^{*}\right)}^{2}+{\left(\Delta b^{*}\right)}^{2}}$$
where Δ*L**, Δ*a**, Δ*b** are the variation parameters of the film respecting its equivalent control sample extract free.

On the other hand, given that the opacity of the packaging film is directly related to its transmittance, a piece of film was placed inside a quartz cuvette and its transmittance was scanned from 200 to 800 nm wavelength using a UV-Vis spectrophotometer (Cary 60 UV-Vis de Agilent Technologies (Santa Clara, CA, USA)). An empty quartz cuvette was used as the control sample.

### 2.7. Solubility and Water Vapor Permeability (WVP) of the Films

The transmission of water vapor between the film and the environment can increase the shelf life of the product and reduce the growth of microorganisms. On the other hand, high solubility could lead to film deformation of the film in an aqueous medium [47], which would also be undesirable in the case of foods with high water activity. Both of these parameters have been determined for the selected films after their antioxidant capacity had been determined.

In order to measure the film’s solubility, the method of Zhang et al. and Hafsa et al. [13,47] was followed, with some modifications. Squares of 1.5 × 1.5 cm film pieces were cut out and placed inside a desiccator until they reached a constant mass value. Then, they were submerged into 50 mL of distilled water for 24 h under agitation at 100 rpm on a magnetic plate in order to evaluate their dissolution in the medium. Then, the films were removed from the medium and kept inside the desiccator again, until their final mass value was constant. Each film test was replicated. Their percentage of solubility (%*S*) was calculated according to Equation (8):(8)$$\%S=\frac{m_{i}-m_{f}}{m_{i}}\times 100$$
where *m_i_* is the initial mass of the film piece and *m_f_* its final mass after the test.

The permeability of the films to water was determined by gravimetry following the ASTM E96-95 gravimetric method (1995). An amount of 5 mL of distilled water was added to glass vials, which were airtight sealed using the active films. The closure was secured using Parafilm. A control film with no MLE added and a vial with no film were also analyzed. The flasks were kept at room temperature inside a silica desiccator and weighed daily for a 15-day period. The analyses were conducted in duplicate. The permeability of the film was represented as percentage (%*P*), as shown in Equation (9):(9)$$\%P=\frac{m_{i}-m_{f}}{m_{i}}\times 100$$
where *m_i_* is the mass of the film measured on day 0 and *m_f_* its mass measured after a given time.

### 2.8. Scanning Electron Microscopy

In order to examine the structure of the films, as well as to detect the possible interactions between the polymers and the MLE, the samples were observed through a Nova NanoSEM 450 Scanning Electron Microscope after its covering with a 10 nm gold layer and a voltage of 5 kV as described by Cejudo et al. [48].

### 2.9. Study of the Migration of Volatile Organic Compounds (VOCs) from the Films

Headspace-Gas Chromatography-Ion Mobility Spectrometry (HS-GC-IMS) was used to evaluate the possible migration of VOCs from the different films. The conditions were set considering previous analysis [49]. The film samples, with and without MLE, as well as a pure MLE sample, were analyzed by Headspace-Gas Chromatography-Ion Mobility Spectrometry (HS-GC-IMS). A set of 2 × 2 cm pieces of films of 0.8 g were placed into 10-mL glass vials and directly analyzed by a FlavourSpec HS-GC-IMS system (G.A.S., Dortmund, Germany) without any pre-treatment of the samples. The HS conditions for the analysis were as follows: 5 min incubation time at 46.3 °C with agitation at 750 rpm. An amount of 100 µL of HS was injected by the syringe which was kept at 51.3 °C. A 20 cm multicapillary gas column MCC OV-5 (G.A.S., Dortmund, Germany) was used. The column temperature was set at 55 °C. The EPC1 (drift gas) was set up at top flow (250 mL/min) in order to prevent the appearance of noise from the non-ionized compounds during the analysis. The EPC2 (carrier gas) was set according to the following ramp: 2 mL/min (t = 0 min), 10 mL/min (t = 5 min), and 25 mL/min (t = 10 min). The total time of the analysis was 15 min. An amount of 3H Tritium beta radiation was used as the ionization source. A total of 99.999% pure nitrogen produced by using a nitrogen generator (G.A.S., Dortmund, Germany) was employed as the drift and carrier gases. A blank was analyzed after each sample to avoid any carryover effect. All the samples were analyzed in duplicate. Two-dimensional topographic plots of the VOCs and the characteristic fingerprint for each sample were obtained by using a Reporter and Gallery plot plug-ins (LAV, G.A.S., Dortmund, Germany). In order to avoid saturation of the equipment signal, the MLE was diluted at 1:500.

### 2.10. Preservation of Raspberry Using Bioactive Films

After determining the best film formulation, its preservative capacity for highly perishable food such as raspberry fruits was assessed. For this purpose, the polymerization of larger size films was required; therefore, a number of films were produced on 22.6 × 33 cm Teflon trays following the protocols specified above. A 240 mL solution was used to maintain the same ratio between solution volume, thickness, and area as in the films previously analyzed. The solution was allowed to dry for 4 days at room temperature. Once the solution had polymerized, the resulting films were cut into 8 × 16 cm rectangular pieces. Each piece was folded in half to create two 8 × 8 cm square envelopes. Three fully mature and fresh units of raspberry were placed inside each envelope. The packages were sealed using transparent adhesive tape and stored at 8 °C for 13 days in order to visually determine the deterioration of the raspberries. The percentage of spoiled raspberries was calculated according to the ratio of visibly deteriorated raspberries respecting the total number of raspberries packaged on each condition.

### 2.11. Statistical Analysis

The results were statistically analyzed using the STATGRAPHICS Plus 4.0 software (Warrenton, VA, USA). A Univariate Analyses of Variance (LSD and ANOVA, *p* < 0.05) was selected to evaluate the significant variables and factors where required.

## 3. Results and Discussion

### 3.1. MLE Production by Enhanced Solvent Extraction

The use of green extraction techniques such as ESE offers some advantages over other ones for the recovery of phenolic compounds. First, this technique minimizes the amount of solvent used, which provides high-concentrated extracts that do not require concentration steps, reducing production costs [50]. Second, the inert atmosphere generates supercritical CO_2_, which avoids the oxidation of compounds. Third, it promotes the extraction of lipidic and other low-polar compounds that can facilitate plasticizing effect in the film-forming process, and finally, the low temperature used in the process avoids the loss of bioactivity of the extracts [51]. Taking all these benefits into consideration, ESE was employed to obtain a high bioactive MLE. The antioxidant and antimicrobial properties of MLE obtained by ESE have been previously reported in previous studies by the authors [27,35], and the bioactivity of the present extracts was within the same range as that determined in the present study (*IC*_50_ of 9.84 μg/mL extract and an AAI of 2.35 μg DPPH/μg). Respecting the antimicrobial properties, the minimal inhibitory concentration (MIC) for *E. coli* and *S. aureus* was achieved at 200 μg/mL in both cases. According to the classification of the extract proposed by Oliveira et al. [52] attending to the MIC value, the MLE obtained can be classified as having strong antimicrobial capacity, which would be in line with the bioactivity previously reported for other extracts obtained using this raw material by supercritical techniques [27,53].

### 3.2. Manufacturing of Biodegradable Active Films

Films were prepared with varying amounts of extract and filmogenic solutions following the formulations indicated in Section 2.3. Table 1 includes the thickness of the films obtained through each formulation together with its extract concentration, which was calculated considering the final weight of the films.

Film thickness can be a determinant of its mechanical strength, extract holding capacity, water vapor permeability, and transparency [35,36]. First, the films polymerized with any extract addition were analyzed. Compared to other films commonly used for food packaging, such as low-density polyethylene (LDPE) or high-density polyethylene (HDPE), chitosan may lead to a higher tensile strength, although this may vary depending on the processing method [54]. However, the 5 mL films were too thin and possess poor polymerization; therefore, they were discarded for further analysis. On the other hand, the starch films presented a gelatinous, plastic texture, which imparts lack of mechanical resistance, which agrees with other authors [55]. In fact, the films made with 10 mL were discarded as they resulted in thick films with reliefs due to irregular drying. This higher moisture level causes an increase in the film weight and therefore a relative lower extract concentration per gram of film, compared to the other two formulations. Apart from those cases, the rest of the polymer formulation exhibited a correct polymerization and an apparent strength. The combination of both polymers, chitosan and starch, resulted in films with an appearance similar to that of chitosan to the naked eye, since probably this polymer has a predominant influence with regard to mechanical resistance. ANOVA tests were performed for thickness as a function of films and no significant differences were obtained (p_CH_ = 0.078; p_ST_ = 0.086; p_CH-ST_ = 0.069).

Xu et al. [56] and Bourtoom and Bourtoom [57] concluded that the combined CH-ST films are more resistant to traction than those made from only one of these polymers. They also confirmed that there is a reaction between the hydroxyl groups (−OH) in the ST polymers and the amino groups (−NH_2_) in CH. From the point of view of the MLE concentration, the volume of filmogenic solution used on each of the compositions does not affect the final weight of the film, and the final concentration of extract calculated was not substantially varied on the three film formulations. However, as can be seen in Figure 1, the addition of MLE alters the color of the films, turning them darker, thicker, and stiffer as the extract concentration increases. The physicochemical effects from the addition of phenolic extracts to chitosan and starch matrices have been previously analyzed in several studies. Piñeros-Hernández et al. [38] studied the interaction of rosemary extract with starch films and concluded that the extract reacted with the hydroxyl groups of starch, reducing their availability to interact with the glycerol used as plasticizer. This resulted in a poorer mechanical strength of the films, which became more brittle as the percentage of rosemary extract increased. On the other hand, the mechanical strength of the chitosan-formed films was benefited by the addition of extracts according to Souza et al. [58], who studied the effect of five essential oils and six hydroalcoholic extracts. Therefore, the results observed in the present study seem to agree with the results reported by the aforementioned authors, even though the mechanical properties of the films should be specifically tested for a solid confirmation of this point.

It can be seen from Figure 1 that the films have a homogeneous and translucent appearance with an orangey color, whose intensity increases as the percentage of MLE and the volume of the filmogenic solution are increased. The analyses of the color properties of the films can be observed in Figure 2 and Appendix A.

Most of samples are located within the first quadrant because of the substantial increment in the yellow/red tonalities represented by the positive *a** and *b** values. As the %MLE was increased, a decrease in the *L** values could be observed in all cases. The color of the films was rather customizable according to their MLE content. Although the MLE extract on its own has a green color, the tight reaction that takes place with the filmogenic solution during the casting process changes it into different amber tonalities. This behavior has also been observed when casting films using nanofibrillated cellulose and MLE, in contrast with the greenish appearance exhibited by the films processed through supercritical impregnation [27]. Although all the films showed the same trend, some differences could be observed depending on the polymer type. Thus, the color of the starch films exhibited more reddish tones than the chitosan films, which still displayed some greenish hues (Appendix A). In addition, chitosan makes a greater contribution to color than starch, as the color profile of the chitosan MLE films remained more invariable and similar to that of chitosan on its own than the color profile of starch MLE films, which exhibited more noticeable alterations. Attending to the Δ*E* index, the effect resulting from the addition of the extract at any of the concentrations used was highly noticeable. As expected, the higher the extract concentration, the greater the color difference between the film and its corresponding control sample. On the other hand, greater volumes of the filmogenic solution resulted in evident color differences in the case of CH and CH-ST films (Δ*E* > 10), but no perceptible color differences were detected between the ST films manufactured using either 5 or 7.5 mL (Δ*E* = 0.58). Other authors, such as Hafsa et al., (2016) [13] have reported similar results about films made from polysaccharides by casting.

Considering these results, the manufacturers could produce films with interesting bioactive properties while keeping the transparency and color of films at desirable levels from the consumers’ point of view.

### 3.3. Bioactive Properties of MLE Films

The releasing of active compounds from films depends largely on the characteristics of both the polymer and the extract, as well as on the characteristics of the medium [59]. In fact, the volume of the polymer solution used for the manufacturing of the film can also be a key factor on the kinetic diffusion of the active compounds [60], being possibly hindered because of a more compact polymeric net. Thus, by monitoring the antioxidant capacity along the very initial hours, when the kinetic diffusion is higher [61], the aforementioned factors regarding the diffusion of MLE can be closely determined.

Figure 3 shows the results obtained. As expected, by increasing the percentage of extract, a higher antioxidant capacity of all the films was generally observed, with the highest levels being achieved when 5% of MLE was added. Nevertheless, in some cases, as CH films, any remarkable differences in using 3% MLE instead were observed in the time studied.

The effect of the amount of filmogenic solution volume used, within the same polymer type, seems to be a lesser factor with regard to MLE release. The only relevant differences were related to low %MLE in starch films, observing higher MLE release on the 5 mL films than from the 7.5 mL ones.

On the other hand, the type of polymer seemed to be determinant with regard to the kinetic release of the extract, which in turn resulted in significant differences in their antioxidant activity levels. The releasing of the antioxidant compounds from ST films was greater and at a faster rate, obtaining a total DPPH reduction in the 5%MLE assays regardless of the filmogenic solution volume used (5 or 7.5 mL). It should be noted that the addition of plasticizers—as MLE can actually act—improve the flexibility of starch films by weakening the hydrogen bonds in their amylose and amylopectin chains [62], which in turn, may promote migration. On the other hand, the chitosan films showed a slower and lesser releasing of the extract, as it was less affected by the MLE addition, due to the presence of amino-charged groups promoting attraction forces between the chains [63], hindering the extract’s release. We must bear in mind that chitosan films, either on their own or combined with ST, exhibit a considerable antioxidant capacity that may reach inhibition levels at around 6% over the reaction time studied, regardless of the volume used (Figure 4a,c). Therefore, in the case of blended films—CH-ST-MLE (Figure 4c)—although the kinetic reaction was governed by the presence of CH rather than by that of ST, they exhibited a slightly lower antioxidant activity than that of pure CH films. Talón et al., had observed the same behavior by CH-ST films when thymol extract was added to them, attributed to the lesser encapsulating capacity of the composed polymers [64].

In view of the results obtained from these analyses, the films selected for a further and deeper study on the rest of the properties were those produced using 7.5 mL polymer solution and 3% of MLE. This agrees with other studies, such as the one by Yang et al. [65], where the amount of thymol and carvacrol added was less than 10% in order to preserve the mechanical properties of a PLA-PBSA film.

The antimicrobial activity of the MLE against *E. coli* and *S. aureus* have already been previously reported by other authors [66,67]. However, this bioactivity is altered when incorporated into active polymers, suffering a certain decrease due to the lower accessibility of the extract to the culture media [27,68]. Table 2 shows the antimicrobial capacity of those films against *E. coli* and *S. aureus*. As expected, the high antimicrobial capacity of CH films stood out from the rest of the films tested, which was evidenced both in the films composed exclusively of this polymer and the blend. The effect of the extract was evident in all the film formulations, but the most significant differences with respect to the control films corresponded to CH and CH-ST films, where the addition of MLE promoted the antimicrobial effect of chitosan on its own. On the other hand, the specific antimicrobial efficiency of MLE against these two microorganisms had already been reported by other studies in nanofibrillated cellulose-casting-films. The inhibition percentages registered were lower than those obtained in this study, which seems to indicate that bioactivity is enhanced when chitosan is used for the film formulation [27]. The effect on the MLE in the ST films is not so clear, since although their inhibition capacity was higher in the presence of the extract, the ST control films on their own exhibited a certain level of inhibition itself. It could be possible that the high solubility of the ST films in aqueous media (Figure 4a) could interfere with the absorbance measurement, hindering the antimicrobial effect of MLE.

### 3.4. Physical Properties

Starch is a polymer of hydrophilic nature due to the abundance of hydroxyl groups (−OH) in its amylose and amylopectin components. These groups have the capacity to retain water molecules by means of hydrogen bonds, which explains the high solubility of this matrix (Figure 4a). In contrast, the acetyl groups (−COCH_3_) along the chitosan chains endow this polymer with a hydrophobic character that is evident in both the pure CH films and the blended CH-ST films [69]. Yu et al., reported a reduction in moisture content and water solubility in hydroxypropyl methylcellulose (HPMC) and hydroxypropyl starch (HPS) film reinforced with chitosan nanoparticles attributed to the decrease in free volume on the polymeric matrix [70], which explains the behavior observed in the CH-ST films.

As already mentioned, active films are intended to be used with highly perishable foods that are not suitable for other preservative treatments. This kind of food, such as fruit or vegetables, is sometimes packed after being peeled or cut into pieces, which results in the exudation of a considerable amount of water that may affect the mechanical resistance of the packaging material.

Although the addition of the extract had a filling effect that reduced the solubility of this polymer in all the cases, the high solubility of the ST films was still rather noticeable, which can be a considerable limitation with regard to their practical application of ST as a food packaging film [9]. Based on all of the above said, together with the description of its functional properties in Section 3.3, this formulation was discarded as inadequate for the food packaging-intended purposes of our study.

The results on solubility match with the water permeability assays (Figure 4b), where the decrease in WVP was similarly enhanced either by the mixture of the polymers or by the addition of the extract. This behavior had already been observed by Hernández et al. [11] who attributed this reduction in WVP to a greater stiffness of the film and an increased hydrophobicity resulting from the addition of the extract. This is a factor to be considered when it comes to fresh food packaging, since a greater water permeability of the film may, on the one hand, prevent the loss of moisture from the packed fruit or vegetables, but on the other hand, this higher moisture level may lead to the growth of molds.

Light barrier is another parameter of importance for the preservation of oxidation-sensitive foods. As can be seen in Figure 4c, the control films proved to be a good barrier against the short wavelengths (λ < 320 nm) of UV light, with zero transmittance within that spectral region. In a study by Bangyekan et al. [71], it was reported that chitosan films exhibited higher transmittance than starch films, which agrees with the results obtained from our assays, where the CH-ST films offered a more efficient barrier against UV and against the visible light spectrum than the CH films. In the case of the control films—without MLE—their differences regarding the transmittance of light became smaller at the higher end of the visible spectral region, where the two formulations presented rather similar transmittance levels. In the case of MLE-added films, this protection extended to the end of the UV region (400 nm) in both formulations. As expected, the extract, by darkening the films, reduced their transmittance, improving blocking throughout the UV-Vis range. This agrees with the data reported by other authors regarding the development of edible chitosan film with the addition of grape seed extract added [72] and curcumin extract [47]. Therefore, biodegradable MLE-added CH or CH-ST films have the capacity to absorb UV radiation at a better rate than their respective controls, and therefore they can prevent the oxidation of photosensitive foods [73]. Even so, both control and MLE-added films showed lower protection in the visible region than in the ultraviolet one.

### 3.5. Scanning Electron Microscopy (SEM)

The SEM images of all the samples were taken at 2500× magnifications (Figure 5). The CH films displayed a more homogeneous surface than the mixed polymer films, although both of them showed good miscibility, without fissures or creases. However, the addition of the MLE significantly modified the morphology of the films. The CH-MLE films presented some lumps, which have already been observed in other matrices, probably due to the presence of MLE [35,48,74]. This effect was more pronounced in the CH-ST-MLE films, where a very large number of creases and protrusions could be observed. The MLE seemed to cause the shrinking of the polymeric mixture, which can be related to a high cross-linking density [75]. Olewnik-Kruszkowska et al. also observed the irregular surface presented by polylactide-polyethyleneglycol (PLA-PEG) films when polymerized with propolis extract [76].

### 3.6. VOCs Migration

The migration of VOCs in relation to food packaging techniques is a topic of growing interest in the industry, regarding the bioactive properties of the VOCs and their capacity to penetrate the food matrix with ease. In this sense, although VOC migration can be considered a positive factor as it may improve food preservation, it may also alter the food composition and, therefore, its quality and organoleptic characteristics. In fact, the use of active packaging with volatile compounds can even have a positive impact by making the food look more attractive for consumers or by improving the food’s flavor/aroma [67]. Therefore, to better understand the mechanism of action involved in the preservation of the food, the migration of VOCs from the most suitable active films (7.5 mL CH and CH-ST films with 3% MLE) was determined. The fingerprint of the extract and the control films (without MLE addition) were also obtained for comparison purposes (Figure 6). The compounds were not identified individually, since the objective was to evaluate the general migration and the dissimilarities of the different film’s behavior, i.e., to present in a simple and convenient manner the volatile profile differences between the films. That is why HS-GC-IMS was used, as this is a rapid and ecofriendly technique that does not require any sample preparation. This analysis technique provides the characteristic volatile profile of each sample based on the VOCs found in the headspace (HS); therefore, the migration of VOCs from the different films can be determined and compared based on the VOCs detected. HS-GC-IMS uses retention times (seconds on the *y*-axis) and a specific drift time (milliseconds relative to Reactant Ion Peak (RIP) on the *x*-axis) to characterize VOCs according to their interaction with the chromatographic column and their separation in the drift tube, respectively. As can be seen in Figure 6, most of the VOCs are eluted before 400 s and produced signals mainly between 1.0 and 1.8 ms (RIP relative). However, as expected, the volatile profiles obtained for the different types of active films, as well as for their respective control films, were different. These differences in the VOC profiles of the films are not exclusively attributable to the presence/absence of the MLE, but also to the nature of the polymer used. These results agree with those reported in a previous review by Lucera et al. in which different types of active packing with VOCs were evaluated based on the packaging materials (polymeric and non-polymeric films) [77]. In this study, the authors also reported different strategies to release the VOCs into the active packaging (either by incorporating them into the polymeric film matrix and/or by coating the film surface).

For a better evaluation and comparison of the migration of VOCs, the characteristic fingerprint of each film and the MLE was obtained (Figure 7). For this purpose, the areas of the most characteristic signals of the extract (higher intensity) at different drift times were selected. As can be seen in Figure 7, signals 1–3 are present in the extract (MLE) as well as in the MLE-added films. However, they do not appear in the control films, which indicates that these MLE compounds had been released from the films into the headspace. It can also be observed that the areas, and therefore the amount of VOCs, were different depending on the type of film. Signal 4 was present in the extract, although at a lower concentration than the rest of the compounds. However, this compound could also be observed with greater intensity in the extract-added films and particularly in the MLE-CH-ST films. These results also reveal that the migration of the different compounds may also vary depending on the type of film, since signals 1 and 3 appear with a greater intensity in the chitosan films (CH 3%MLE) while compounds 2 and 4 displayed a higher intensity when starch was added to the polymer formulation (CH-ST 3%MLE).

These results demonstrate that some VOCs from the extract migrate to the headspace, showing different releasing behavior depending on the polymer base, since they registered dissimilar volatile fingerprints. The bioactive properties of the extracts would not be exclusively attributable to their non-volatile phenolic compounds, but also to the VOCs that were released from the active packaging. In a previous study, Avila-Sosa et al. demonstrated the potential of essential oils from oregano, cinnamon, or lemongrass as antifungal agents added to chitosan and starch-edible films. The authors reported that the antifungal efficiency of the films improved when the VOCs were released by vapor contact rather than when there was a direct contact with the food [78].

In this sense, these results could be the starting point for a deeper study targeting the identification and quantification of these active VOCs in order to determine the better packaging strategy to guarantee their contribution to the preservation of the quality and safety of food, their influence on the food organoleptic properties, and the perceived intensity of these VOCs by consumers.

### 3.7. Raspberry Preservation over Storage Time

The chitosan and chitosan/starch film formulations with an addition of 3% MLE were employed in a 13-days preservation experiment of fresh raspberries. The experiment focused on the visual spoilage of the fruit, also monitoring the evolution of the unpacked fruit (Figure 8). The unpackaged raspberries showed significantly shorter shelf-life than the packed ones, so that just after 3 days, they presented a visually noticeable loss of firmness due to the absence of a water vapor barrier. However, the packaged samples kept their firmness up to the 6th day in all cases, attributable to the low water vapor permeability and solubility of the films reported in Section 3.4.

Considering that raspberries are climacteric fruits, the loss of weight and firmness can be attributable to the respiration rates. As Huynh et al., reported, the loss weight of berries packed into modified atmosphere was lower respective to the one observed when using perforated starch films because of the suppression of the fruit metabolism when using a mix of gases [79].

The preservation of the food firmness by means of active packaging materials had been previously reported by Olewnik-Kruszkowska et al., who demonstrated the successful conservation of blueberries’ firmness when packed in propolis-added PLA-PEG films with respect to their respective control samples after 14 days of storage [76]. Similar results were reported by Sun et al. on the storage of strawberries coated with carboxymethyl-chitosan containing clove oil [80]. In this study, until the 13th day, the action of the MLE on the film’s formulation was not evident, since the preservative effect of chitosan prevailed. On the last day of the tested period, a higher spoilage of the raspberries packed in CH films could be observed. Considering the grey color of the mycelia observed on the raspberry surface, the spoilage can be probably due to the presence of *Botrytis cinerea*, which is one of the main microorganisms responsible for the spoilage of raspberries after harvest [81]. All the fruit packaged into the control CH film showed an absence of firmness and appearance on exude, while the CH-MLE film only showed a 55.5% amount of spoiled raspberries (Table 3). A slight influence of the extract was observed at this point, as a decrease of 10% in spoilage on both film formulations is shown. The higher oxygen permeation of chitosan possibly promoted the mold growth, which decreased when the blend CH-ST was used due to a higher abundance of hydrogen bonding [82], which was even more reduced when the extract was added, agreeing with other authors [83]. A very high loss of gas permeability—both oxygen and carbon dioxide—can act in detriment of the preservation by favoring the anaerobic facultative bacteria growth. However, this effect was probably counteracted by an increment in the number of antioxidant and antimicrobial compounds contributed by the MLE to the packaging. According to the data recorded, the VOCs released by the MLE over the storage period seemed to provide enough active properties to favor a better preservation of the fruit, given that the best preservative efficiency was exhibited by the MLE-added CH-ST polymers.

## 4. Conclusions

The formulation of active polymers for the packaging and preservation of food entails a broad study of the film properties from physical and bioactive perspectives. In this study, some of the factors that may affect the effectiveness of active packaging materials have been studied. For this purpose, CH, ST, and CH-ST films have been added to MLE at different concentrations. Increasing amounts of MLE provided better protection against UV and enhanced antioxidant activity, even if just adding 3% MLE to the different formulations was enough to obtain homogeneous films with good physical integrity and bioactive properties. The ST films released large amounts of MLE into the DPPH reagent, while the CH and CH-ST films tended to retain the MLE compounds, decreasing its release. Nevertheless, the antimicrobial action on the liquid culture media was substantially greater in the CH and CH-ST films because the endogenous antimicrobial effect of this agent promotes the bioactivity of the MLE. So, all in all, the bioactivity of the polymers was guaranteed.

The composition of the food to be preserved, and particularly its water content, is a factor to be considered in bio-based polymers, since it may affect the film resistance. In fact, the ST films exhibited quite a high solubility in water that could affect their integrity in the long term. Depending on the mode of action of the polymer, its preservative effect can also be developed in the headspace of the package, which would affect packaging requirements. In this sense, the results obtained from the VOC migration study on the selected films (CH 3%MLE and CH-ST 3%MLE) confirmed the presence of the VOCs from the MLE in the headspace of the package. This migration of VOCs from the MLE into the headspace presented different profiles depending on the polymer type used, which may also represent another factor to consider with regard to food preservation. Furthermore, the superior preservation efficiency (over 10%) of the bioactive films compared to that of the non-active ones makes us think that the VOCs released into the HS exert a certain antimicrobial action, given that the film packaging method did not involve the vacuum sealing of the package and therefore, the film was not in tight contact with the food.

This study provides interesting results for the control of influential factors in the manufacture of active packaging with biodegradable films, as well as its potential in food preservation, based on the results obtained in real food. The reduction in waste of highly perishable and high-cost products, such as raspberries, can lead to economic savings for producers and consumers.

## Figures and Tables

**Figure 1 foods-12-02977-f001:**
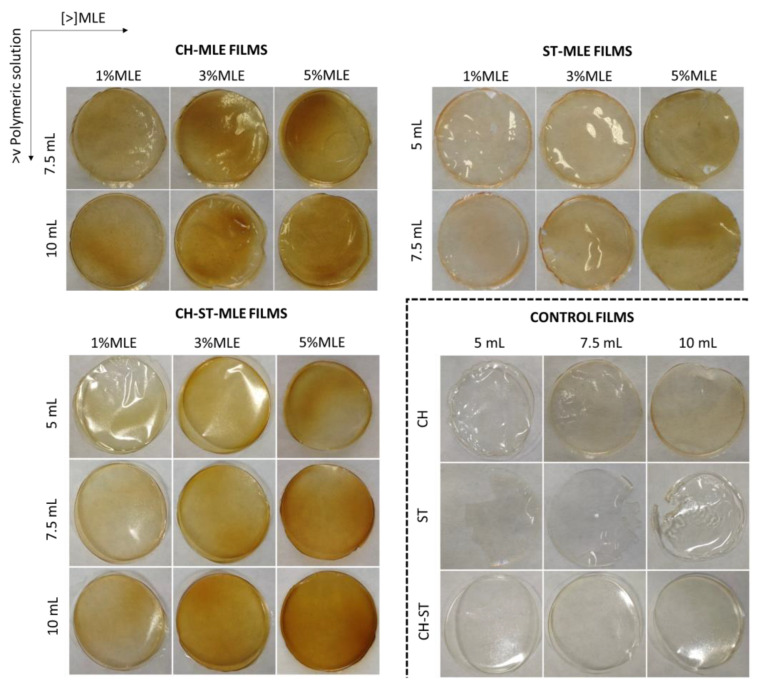
Appearance of the active MLE biopolymers. The images enclosed by the dot line correspond to the control films (without MLE addition).

**Figure 2 foods-12-02977-f002:**
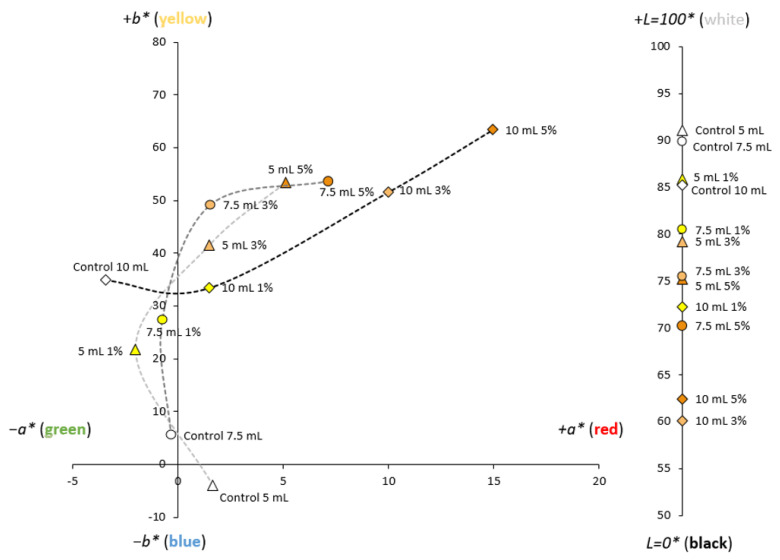
CIELAB coordinates of CH-ST films manufactured using different volumes (Δ: 5 mL; ○: 7.5 mL; ◊: 10 mL) and MLE percentages. The graphics corresponding to the CH and ST films are included in the Appendix A.

**Figure 3 foods-12-02977-f003:**
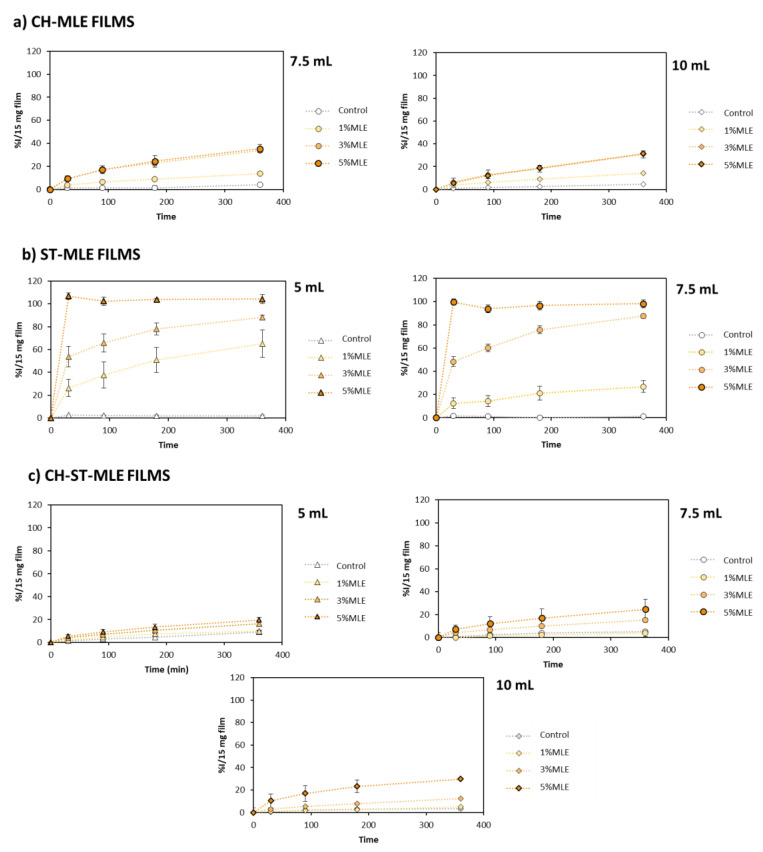
Antioxidant activity of active films manufactured using different volumes (Δ: 5 mL; ○: 7.5 mL; ◊: 10 mL) and MLE percentages.

**Figure 4 foods-12-02977-f004:**
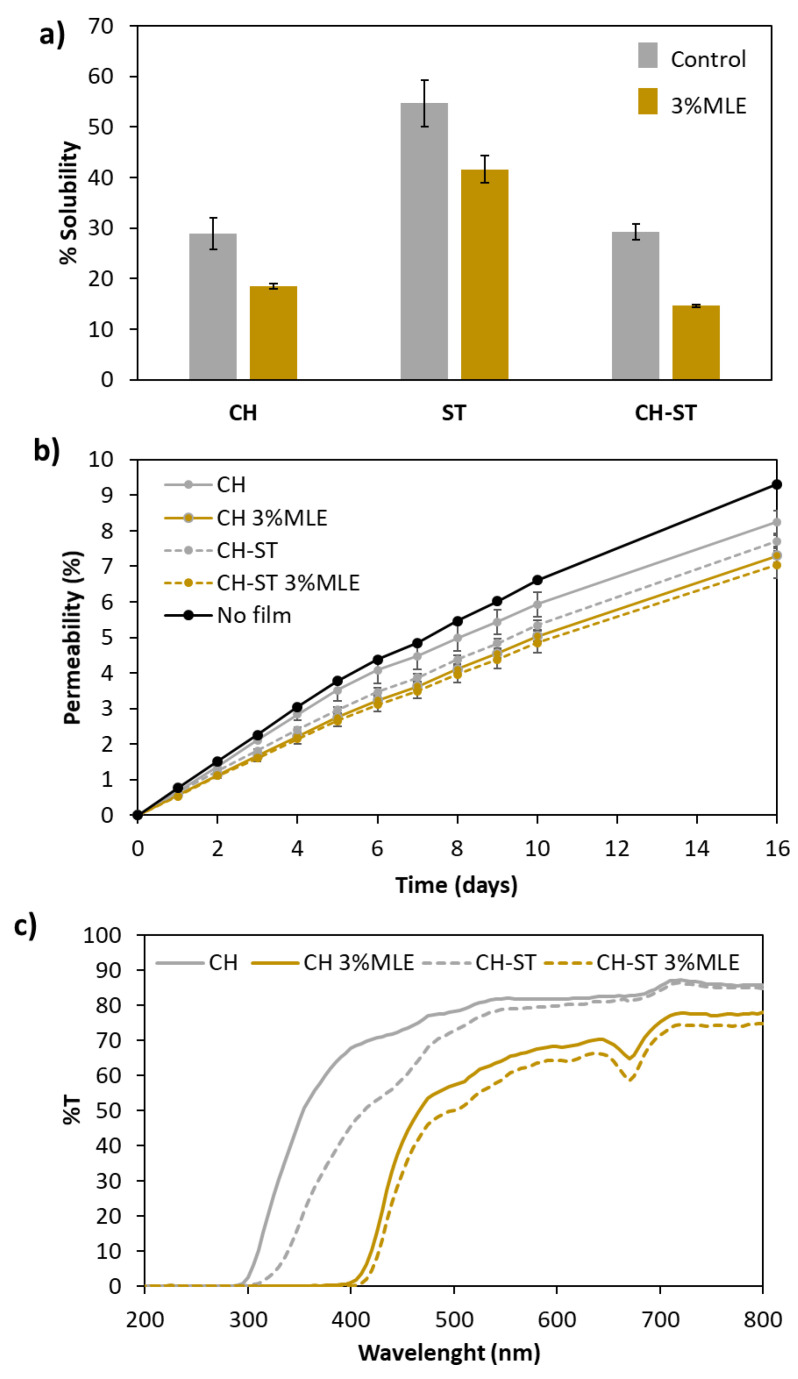
Physical properties of the manufactured films: (**a**) solubility, (**b**) permeability and (**c**) transmittance.

**Figure 5 foods-12-02977-f005:**
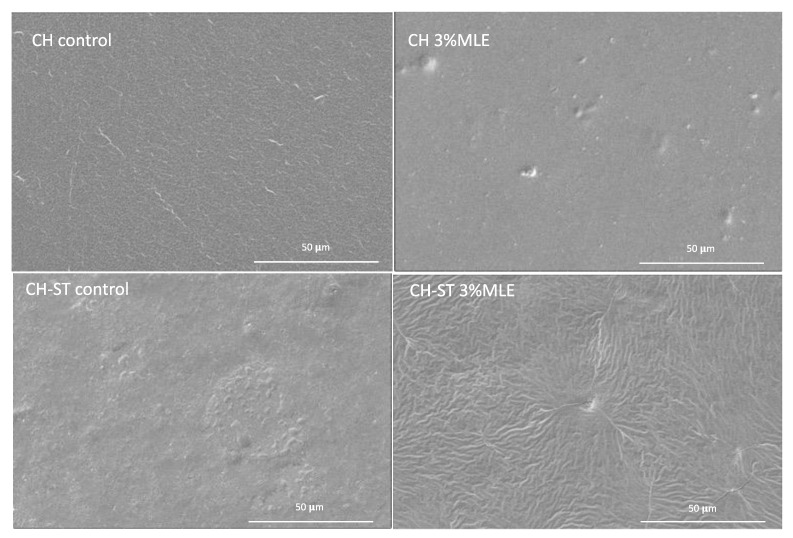
MLE-added and MLE-free film surface images (800×).

**Figure 6 foods-12-02977-f006:**
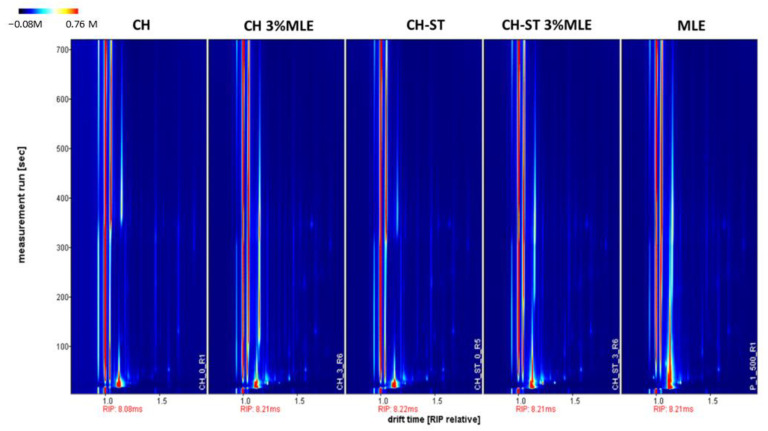
Two-dimensional GC- IMS topographic plots of the selected active films: Chitosan and Chitosan-Starch control of films (CH-0 and CH-ST 0) and Chitosan and Chitosan-Starch with 3% of MLE added (CH-3 and CH_ST 3). The intensity value (V) is indicated by a color scale (low intensity in white; high intensity in red).

**Figure 7 foods-12-02977-f007:**
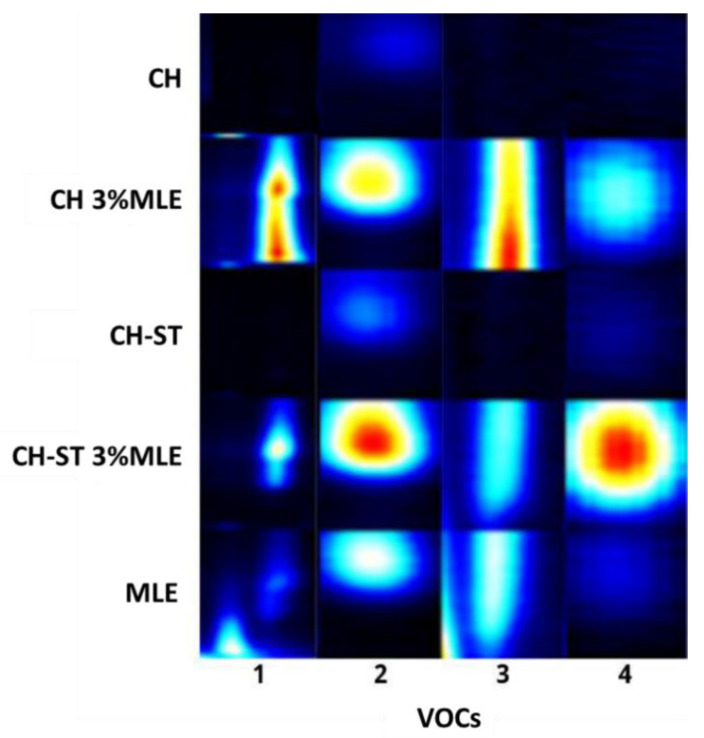
Characteristic fingerprints obtained by HS-GC-IMS for each of the selected active films and the MLE.

**Figure 8 foods-12-02977-f008:**
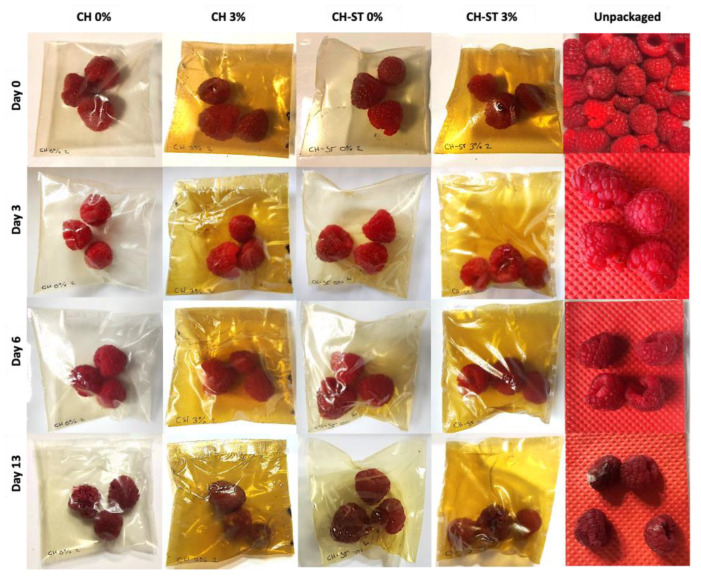
Packaging assay of raspberry fruit.

**Table 1 foods-12-02977-t001:** Thickness values and composition of the films performed at the different formulations (*n* = 2).

		Thickness (mm)			mg MLE/g Film		
Volume of Film Solution	MLE Addition (% *v*/*v*)	CH	ST	CH-ST	CH	ST	CH-ST
5 mL	Control	-	0.12 ± 0.01	0.05 ± 0.01	-	-	-
	1%MLE	-	0.13 ± 0.02	0.04 ± 0.01	-	8.99 ± 0.16	36.28 ± 0.44
	3%MLE	-	0.10 ± nd	0.11 ± 0.03	-	34.60 ± 0.08	101.65 ± 1.75
	5%MLE	-	0.13 ± nd	0.13 ± 0.02	-	44.72 ± nd	160.85 ± 1.15
7.5 mL	Control	0.08 ± 0.01	0.23 ± 0.05	0.14 ± 0.01	-	-	-
	1%MLE	0.08 ± 0.01	0.19 ± 0.04	0.11 ± 0.01	35.15 ± 0.07	8.77 ± nd	35.54 ± 0.11
	3%MLE	0.14 ± 0.04	0.19 ± 0.01	0.12 ± 0.02	97.73 ± 3.07	36.25 ± 0.78	105.92 ± 0.57
	5%MLE	0.12 ± 0.04	0.20 ± 0.01	0.11 ± 0.02	141.85 ± 7.53	42.83 ± nd	174.54 ± 2.25
10 mL	Control	0.08 ± 0.01	-	0.06 ± nd	-	-	-
	1%MLE	0.14 ± 0.04	-	0.06 ± nd	35.25 ± 0.18	-	35.12 ± 0.09
	3%MLE	0.12 ± 0.04	-	0.16 ± 0.01	105.31 ± 5.77	-	100.85 ± 0.15
	5%MLE	0.13 ± 0.04	-	0.15 ± 0.02	175.31 ± 2.69	-	168.52 ± 0.78

**Table 2 foods-12-02977-t002:** Antimicrobial capacity of the films.

Film Formulation	*S. aureus* (%*IMG*/50 mg Film)	*E. coli* (%*IMG*/50 mg Film)
CH control	62.83 ± 2.26 ^b^	74.65 ± 3.54 ^c^
CH 3% MLE	76.82 ± 1.02 ^c^	77.55 ± 1.60 ^c^
ST control	11.12 ± 1.30 ^a^	7.28 ± 2.28 ^a^
ST 3% MLE	12.86 ± 3.31 ^a^	8.92 ± 0.08 ^a^
CH-ST control	63.02 ± 2.35 ^b^	64.75 ± 1.42 ^b^
CH-ST 3% MLE	80.13 ± 0.16 ^c^	74.09 ± 3.19 ^c^

The different letters in the columns indicate significant differences. *p* < 0.01 LSD test.

**Table 3 foods-12-02977-t003:** Percentage of raspberry spoiled on preservation assay.

Film Formulation	Day 3	Day 6	Day 13
Control	0/9 (0%)	2/9 (22.2%)	7/9 (77.8%)
CH 0%	0/9 (0%)	0/9 (0%)	5/9 (55.5%)
CH 3%	0/9 (0%)	0/9 (0%)	4/9 (44.4%)
CH-ST 0%	0/9 (0%)	0/9 (0%)	4/9 (44.4%)
CH-ST 3%	0/9 (0%)	0/9 (0%)	3/9 (33.3%)

## Data Availability

The data used to support the findings of this study can be made available by the corresponding author upon request.

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
