# Peer review of "Functional, Physical, and Volatile Characterization of Chitosan/Starch Food Films Functionalized with Mango Leaf Extract"

_foods, 2023, doi:10.3390/foods12152977_

Round 1

Reviewer 1 Report (Previous Reviewer 2)

There are minor issues that should be addressed by the authors:

In abstract, the full name of HS GC-IMS should be provided.

Ln 131, the citation should be Siripatrawan and Harte [36], also ln 141 the citation should be Reddy and Yang [39].

Ln 197, 1.5*106 CFU/mL, correct, also in line 219.

All equations should be supported with the relevant references.

Ln 220, UFC/mL, revise

Ln 255, water activity (aw), there is no further use for this abbreviation, I propose deleting it.

Ln 408, Esposito et al. (2020), the ref. no. is missing.

Ln 579, RIP, identify.

PLA-PEG films, identify at the first appearance.

The conclusion section is overly lengthy.

Author Response

Authors appreciate the evaluation of the reviewers, which is focused on improving the quality of the manuscript. All suggestions have been amended as can be seen highlighted in green in the manuscript, and the queries have been responded to one by one.

Revisor 1

In abstract, the full name of HS GC-IMS should be provided. Amended

Ln 131, the citation should be Siripatrawan and Harte [36], also ln 141 the citation should be Reddy and Yang [39]. Amended

Ln 197, 1.5*106 CFU/mL, correct, also in line 219. Amended

All equations should be supported with the relevant references.

Response: Please kindly note that some changes has been done on the material and method section. All analysis indicate the methodology employed, where the equations are also described.

Ln 220, UFC/mL, revise Amended

Ln 255, water activity (aw), there is no further use for this abbreviation, I propose deleting it. Amended

Ln 408, Esposito et al. (2020), the ref. no. is missing. Amended.

Response: That reference has been removed from line 412.

Ln 579, RIP, identify. Amended.

PLA-PEG films, identify at the first appearance. Amended.

The conclusion section is overly lengthy. Amended.

Response: Please kindly note that in previous revisions it has been asked to highlight the novelty of the study, and one of the most innovative points is the VOC study. In order to not jeopardized that revision, only the most general sentences have been removed. As can be seen, the following paragraphs have been removed.

From line 689.

“Furthermore, the migration of bioactive compounds and their preservative efficiency might be compromised into a polymeric matrix, so the release from the films must be tested on every formulation”.

From line 719 onwards.

“In addition, the preliminary results obtained from the VOCs migration from different biofilms could be considered as the first steps towards a new research line for the quality control of active films manufacture.” This has been removed because is also commented ad the end of the VOC section

In a subsequent study, the migrating compounds should be identified in order to more precisely determine the antioxidant and antimicrobial potential of the headspace as well as to ascertain the effect of these VOCs on the organoleptic characteristics of the packed food”.

Reviewer 2 Report (New Reviewer)

This manuscript describes a topic of great interest for food preservation. However, I have the following observations.

ABSTRACT

In the abstract, the objective of the research must be added.

KEYWORDS

·      The keywords must be different from the words mentioned in the title. In the case of the keyword "mango leaf extract", it could be replaced by the scientific name (Mangifera indica).

·      I also suggest not using abbreviations and putting complete sentences, for example Volatile Organic Compounds.

Introduction

·      The author must specify that the fresh products are vegetables and fruit because it is not clear at the beginning of the introduction.

·      The author must explain why he used raspberry to evaluate.

Materials and Methods

·      The author must indicate the variety of mango used for the extracts

·      ¿What is the degree of deacetylation of chitosan used?

·      ¿Were pathogenic or attenuated strains of bacteria used?

Results and Discussion

·      Shelf life of raspberries: ¿How was Botritys cinerea identified in the fruits?

·      In the discussion, the mechanism of action by which chitosan and starch films reduce water loss and how it intervenes in the regulation of gases in raspberry fruits should be integrated.

Author Response

This manuscript describes a topic of great interest for food preservation. However, I have the following observations.

 The authors appreciate the evaluation of the reviewers. All suggestions have been amended and can be seen highlighted in blue in the manuscript. All the queries have been answered below one by one.

ABSTRACT

In the abstract, the objective of the research must be added.

Response: The authors agree with the suggestion. In order not to exceed the 200-words format, and not remove valuable information, just a small sentence has been included on line 18. It can be read: “to evaluate their food preservation efficiency”

KEYWORDS

  • The keywords must be different from the words mentioned in the title. In the case of the keyword "mango leaf extract", it could be replaced by the scientific name (Mangifera indica). Amended
  • I also suggest not using abbreviations and putting complete sentences, for example Volatile Organic Compounds. Amended

Introduction

  • The author must specify that the fresh products are vegetables and fruit because it is not clear at the beginning of the introduction.

Response: Please kindly note that the sentence of line 37 has been modified. It can be read: “as fourth-range products based on fruit and vegetables”. Besides, on line 71 can be reed again the topic is packed fruits or vegetables.

  • The author must explain why he used raspberry to evaluate.

Response: Please kindly note that the at the end of the introduction section, on line 102, can be read “The selected films were subsequently used to preserve raspberries as an example of a highly-perishable fresh product.”

Materials and Methods

  • The author must indicate the variety of mango used for the extracts.

Response: Please kindly note that the variety of the mango plant was already mentioned on line 107. It is Kent variety.

  • ¿What is the degree of deacetylation of chitosan used?

Response: Please kindly note that the deacetylation degree of chitosan was already mentioned on line 112. It has been highlighted in blue.

  • ¿Were pathogenic or attenuated strains of bacteria used?

 Response: The strains were pathogenic bacteria. That information has been included on line 119.

Results and Discussion

  • Shelf life of raspberries: ¿How was Botritys cinerea identified in the fruits? Response: Authors agree with the observation. Please kindly note that the sentence was modified in line 652, including that information. It can be read: “Considering the grey color of the mycelia observed on the raspberry surface…”

  • In the discussion, the mechanism of action by which chitosan and starch films reduce water loss and how it intervenes in the regulation of gases in raspberry fruits should be integrated.

Response: The regulation of gases has not been tested by the authors experimentally, just the water permeability, which results have been already related to raspberry preservation (line 641 highlighted in blue). However, some references about the gas permeability of starch and chitosan have been also included in section 3.7, aimed at improving the discussion.

Reviewer 3 Report (New Reviewer)

The manuscript describes the functional, physical, and volatile properties of chitosan-starch food films functionalized with mango leave extract. The topic is interesting; However, the manuscript has several problems:

1. Express the type of starch in the topic and other parts of the manuscript.

2. L 22; Specify the storage period.

3. Please elaborate on the functional properties of chitosan-based films according to 10.1016/j.carbpol.2023.120901. 

4. L 106; "L." no need to be in italics.

5. L 109; CO2.

6. L 112, What kind of starch? Provide some more details, such as amylase content. 

7. L 163; All methods should be mentioned and described in the Materials and Methods Section. Also, this paragraph does not need any reference.

8. Part 2.8; Express the magnification.

9. At the end of the Materials and Methods Section, the "Statistical Analysis" section should be added.

10. All Tables and Chart bars should be statistically analyzed with the significant letters, and "a" should be utilized for the highest value.

Author Response

Authors appreciate the evaluation of the reviewers, which is focused on improving the quality of the manuscript. All suggestions have been amended as can be seen highlighted in green in the manuscript, and the queries have been responded to one by one.

The manuscript describes the functional, physical, and volatile properties of chitosan-starch food films functionalized with mango leave extract. The topic is interesting; However, the manuscript has several problems:

  1. Express the type of starch in the topic and other parts of the manuscript. Amended. The information has been included in section 2.1. where all reagents are described.
  2. L 22; Specify the storage period. Amended
  3. Please elaborate on the functional properties of chitosan-based films according to 10.1016/j.carbpol.2023.120901. 

Response: In previous revisions, it has been asked to explain in more detail the analysis carried out. Although is more convenient sometimes to shorten the methodology, it would be in contradiction with the previous revisions, so unfortunately, we have to decline the suggestion.

  1. L 106; "L." no need to be in italics. Amended.
  2. L 109; CO2. Amended.
  3. L 112, What kind of starch? Provide some more details, such as amylase content.

Response: The CAS number of the starch reagent is  9005-84-9. The information about the amylose content has been included in 112 line.

  1. L 163; All methods should be mentioned and described in the Materials and Methods Section. Also, this paragraph does not need any reference.

Amended.

Response: Plese kindly note that some changes has been done according to your suggestions. Ref 41 and 42 have been removed from line 163 as the reviewer suggests. The reference of the methods used has been revised and all of them are highlighted in green. Two new references to the methods have been included.

  1. Part 2.8; Express the magnification. Amended

Response: Authors agree with the suggestion. Please kindly note that the information has been included on the tittle of Figure 5 to a better comparison.

  1. At the end of the Materials and Methods Section, the "Statistical Analysis" section should be added.

Amended.

Response: According to the reviewer suggestions, a new section has been included. It can be seen on line 318: “The results were statistically analyzed using the STATGRAPHICS Plus 4.0 software (Virginia (VA), EE.UU).  A Univariate Analyses Of Variance (LSD and ANOVA, p<0.05) was selected to evaluate the significant variables and factors where required”.

  1. All Tables and Chart bars should be statistically analyzed with the significant letters, and "a" should be utilized for the highest value.

Response: Please kindly note that the statistical analysis has been done in those analyses where it is needed to discuss the results from a statistical point of view, and where it is hard to obtain a conclusion just by attending at the standard deviation. In this sense, the statistical differences have been included in Table 1, the solubility, and in the microbial analysis. However, in Table 1 for thickness an ANOVA has been included but the results show no significant differences.

Regarding the antioxidant capacity, the statistical analysis is not needed and just the standard deviations stand out the differences in the migration behavior. The differences among the data on each of the films are clear, and the statistical analysis in some samples with that high standard deviation is not accurate. The other analysis, as UV spectra, is never represented with a standard deviation.

Round 2

Reviewer 3 Report (New Reviewer)

The manuscript is acceptable.

This manuscript is a resubmission of an earlier submission. The following is a list of the peer review reports and author responses from that submission.

Round 1

Reviewer 1 Report

Based on my pondered analysis, the positive point of this manuscript is the fact that the introduction is well described, the experiment was presented comprehensively, the results are supported by the literature with appropriate citations provided. However I found also weak points, which I have listed below, and encourage the authors to address them.

Major comments:

Materials and methods

1.       What was the pH of filmogenic solutions? The acidity may change the bioactivity of yours samples.

2.       Could you please present the experimental design? In my opinion, authors should performed ratio or amounts of used ingredients with coding in table. It would be much clearer.

3.       Why have not you performed another antioxidant activity method, such as ABTS, αTEAC, FRAP or CL?

4.       Why have not you analyzed statistically your data? Please identify standard deviation at least.

5.       How many production and analytical repetition have you done?

Reviewer 2 Report

I have completed review of “Functional, Physical, and Volatile Characterization of Chitosan/Starch Food Films Functionalized with Mango Leave Extract”, the authors reported original work on the design of food packaging films from chitosan (CH), starch (ST), and mango leaf extract (MLE) different blends. They have examined the optical properties, Water Vapor Permeability, morphology of different formulations as well as the antimicrobial activity against E. coli (Gram-negative) and S. aureus (Gram-positive). Also, the antioxidant capacity was examined by the authors. However, there are some issues that make this manuscript (in the present form) not suitable for publication by Foods@MDPI:

The language should be polished a little bit. There are several instances where the language could be improved for clarity and precision.

The abstract

The abstract part should clearly state the novelty of the work and also include essential information about the methods and characterization of the CH/ST/MLE blends.

Names of microorganisms should be italic. Revise this point in the manuscript thoroughly.

Lns 17-18, “mango leaf extract at varying concentrations (MLE)” change to ““mango leaf extract (MLE) at varying concentrations”

Ln 19, “A CH-ST blend with 3% MLE using 7.5 mL of the polymeric solution” 7.5 mL of the polymeric solution??? Clarify this sentence.

Introduction

The quality of the introduction part should be improved by providing the reader essential information about origin/structure/properties of the biopolymers used in this study. Not limited to these, I propose to use the following recent articles for this purpose.

Applied Biological Chemistry 65(1)(2022) 54, International Journal of Biological Macromolecules 243(2023) 125180, Food Bioscience 44 (2021) 101352, Foods 10 (2021) 1171

Materials and methods

This section should be amended and revised carefully. There are many errors, for example, Ln 115, “91.8 g/mL” how?!!!, In line 130, you prepared 5% starch solution, then in line 135 you blended with 2% ?!!! is this a typo? Ln 156, the concentration 250 twice , Ln 167, revise the 6×10-5 mol/L, revise. Repeated many times in the manuscript. Revise thoroughly.

Degree of deacetylation (DD) of chitosan should be provided because it is one of the parameters which influence the properties of chitosan.

For most of tests, the authors preformed in duplicate. Generally, at least three times parallel test should be performed.

All equations should be supported with the relevant references.

Results and Discussion

Organization of this section is poor and confusing and needs substantial improvement especially for 3.1. MLE production by Enhanced Solvent Extraction (the authors did not discuss the production method under the mentioned title!!!), 3.2. Manufacturing of biodegradable active films (The interaction between the blend formulation components CH/ST/MLE should be described well and the polymerization rection should be representatively presented and confirmed). 3.3. Bioactive properties of MLE-films (poor organization).

Ln 335, CH-PS?!!! also in Figure 1. Control films.

Ln 339, this finding “ST suppressed the crystalline peaks of CH, giving rise to amorphous regions that improve the plasticity of the resulting films” should be supported by XRD characterization.

The change in crystallinity should be approved by XRD.

Ln 482, you have mentioned “The results are in agreement with those obtained by Yu et al. (2023)” but no results were mentioned. You only stated what is reported previously about solubility of ST and hydrophobicity of CH.

The mechanical properties of the obtained blend CH/ST/MLE films should be investigated.

Conclusion

The conclusions are too repetitive and should be improved to well reflect what was achieved from the work hypothesis.

References

Citation way of references in the text is not standardized and wrong in some places. All years should be deleted. For examples Ln 113 the citation should be Rosales et al [29], Ln 119 Siripatrawan and Harte [30] and delete the year, Ln 129 Reddy and Yang [33], Ln 153, ref no. is missing, Lns 218,388,478, etc. All citations should be thoroughly revised.

The language should be polished a little bit. There are several instances where the language could be improved for clarity and precision.